# PARE: A framework for removal of confounding effects from any distance-based dimension reduction method

**Andrew A. Chen**[1]*, **Kelly Clark**[2], **Blake E. Dewey**[3], **Anna DuVal**[3], **Nicole Pellegrini**[3], **Govind Nair**[4], **Youmna Jalkh**[5], **Samar Khalil**[5], **Jon Zurawski**[5], **Peter A. Calabresi**[3], **Daniel S. Reich**[4], **Rohit Bakshi**[5,6], **Haochang Shou**[2,7☯], **Russell T. Shinohara**[2,7☯], **Alzheimer's Disease Neuroimaging Initiative, and North American Imaging in Multiple Sclerosis Cooperative**

1 Department of Public Health Sciences, Medical University of South Carolina, Charleston, South Carolina, United States of America, 2 Penn Statistics in Imaging and Visualization Center, Department of Biostatistics, Epidemiology, and Informatics, University of Pennsylvania, Philadelphia, Pennsylvania, United States of America, 3 Department of Neurology, Johns Hopkins University School of Medicine, Baltimore, Maryland, United States of America, 4 Translational Neuroradiology Section, National Institute of Neurological Disorders and Stroke, National Institutes of Health, Bethesda, Maryland, United States of America, 5 Department of Neurology, Brigham and Women's Hospital, Harvard Medical School, Boston, Masschusetts, United States of America, 6 Department of Radiology, Brigham and Women's Hospital, Harvard Medical School, Boston, Massachusetts, United States of America, 7 Center for Biomedical Image Computing and Analytics, University of Pennsylvania, Philadelphia, Pennsylvania, United States of America

☯ These authors contributed equally to this work.
* chenandr@musc.edu

## Abstract

Dimension reduction tools preserving similarity and graph structure such as *t*-SNE and UMAP can capture complex biological patterns in high-dimensional data. However, these tools typically are not designed to separate effects of interest from unwanted effects due to confounders. We introduce the partial embedding (PARE) framework, which enables removal of confounders from any distance-based dimension reduction method. We then develop partial *t*-SNE and partial UMAP and apply these methods to genomic and neuroimaging data. For lower-dimensional visualization, our results show that the PARE framework can remove batch effects in single-cell sequencing data as well as separate clinical and technical variability in neuroimaging measures. We demonstrate that the PARE framework extends dimension reduction methods to highlight biological patterns of interest while effectively removing confounding effects.

## Author summary

Powerful tools such as *t*-SNE and UMAP can be used to visualize biological patterns in medical data, including brain imaging and gene expression. However, these visuals can be influenced by unwanted patterns. We introduce the partial embedding (PARE) framework, which separates biological patterns from unwanted influences. We then develop partial *t*-SNE and partial UMAP and apply these methods to genomic and neuroimaging

**Data Availability Statement:** The code for PARE are available on Github: https://github.com/andy1764/PARE. The data for Case Study 1 are

openly available and can be downloaded via the scRNAseq package (version 2.7.2) for R, available via https://www.bioconductor.org/packages/release/data/experiment/html/scRNAseq.html. The data for Case Study 2 are openly available as part of the Alzheimer's Disease Neuroimaging Initiative (ADNI), which can be requested via https://adni.loni.usc.edu/ The data for Case Study 3 are not shared publically and must be requested by contacting the North American Imaging in MS Cooperative (NAIMS), https://www.naimscooperative.org/.

**Funding:** This work was supported by the National Institute of Neurological Disorders and Stroke (grant numbers R01 NS085211 and R01 NS060910 and the Intramural Research Program), the National Multiple Sclerosis Society (RG-1707-28586 to R.T.S), the National Institute of Mental Health (R01 MH123550 to R.T.S and R01 MH112847 to R.T.S), and a seed grant from the University of Pennsylvania Center for Biomedical Image Computing and Analytics (CBICA) to H.S. The content is solely the responsibility of the authors and does not necessarily represent the official views of the funding agencies. The funders had no role in study design, data collection and analysis, decision to publish, or preparation of the manuscript.

**Competing interests:** RB has received consulting fees from Bristol-Myers Squibb and EMD Serono and research support from Bristol-Myers Squibb, EMD Serono, and Novartis. DSR has received research funding from Abata Therapeutics, Sanofi-Genzyme, and Vertex Pharmaceuticals, all unrelated to the current study. RTS receives consulting income from Octave Bioscience and compensation for scientific reviewing from the American Medical Association.

data. We first show that the PARE framework can remove technical artifacts in single-cell gene expression data. Then, we show that PAREs can reduce unwanted scanner-related bias in brain imaging data. We demonstrate that the PARE framework highlights biological patterns of interest while effectively removing unwanted effects.

## 1 Background

Dimension reduction tools such as principal coordinates analysis (PCoA), $t$-distributed stochastic neighbor embedding ($t$-SNE), and uniform manifold approximation and projection (UMAP) are widely employed for exploration of high-dimensional data. These methods all identify lower-dimensional embeddings in Euclidean space that preserve information in the original space. These methods have been demonstrated to reveal complex patterns including cell lineages in single-cell RNA sequencing (scRNA-seq) data [1] and neurodevelopmental changes in brain volumetric data [2]. However, in their current form, these methods do not account for covariates and are known to be substantially influenced by confounders such as batch [3].

Researchers have developed several extensions of dimension reduction tools that are designed for removal of confounding effects. For principal component analysis (PCA), researchers developed PCA with adjustment for confounding variation [4]. Adjusted PCoA (aPCoA) examines residuals from a linear model on principal coordinates, which are orthogonal to specified confounding variables [5]. Projected $t$-SNE orthogonalizes the embeddings at each iteration of the $t$-SNE optimization to adjust for batch effects [6]. Another method addresses batch effects by using $t$-SNE to construct a reference embedding based on one batch and then projects observations from other batches onto the reference [7]. To date, adjustment for confounders in distance-based dimension reduction methods has required modification of each framework to address this specific problem. Furthermore, many methods including UMAP have not been extended to address confounding.

We develop the partial embedding (PARE) as a generalizable framework for removing nuisance effects from any distance-based dimension reduction method. We achieve this by using the covariate-adjusted dissimilarities from aPCoA as inputs into dimension reduction methods. When the original distances are Euclidean, we can achieve identical results by treating adjusted principal coordinates as input data (see Methods). We refer to these covariate-adjusted dimension reduction results as partial embeddings (PAREs). These PAREs preserve pairwise distances from the original space while removing confounding effects. PAREs can be produced from a broad class of dimension reduction methods including $t$-SNE [8], UMAP [9], Laplacian Eigenmaps [10], diffusion map embeddings [11], LargeVis [12], TriMap [13], ForceAtlas2 [14] and others. Specifically, we apply the PARE framework to $t$-SNE and UMAP to develop partial $t$-SNE (p-$t$-SNE) and partial UMAP (p-UMAP).

## 2 Methods

Our PARE framework enables removal of confounders from any dimension reduction method that utilizes pairwise dissimilarity values computed between subjects. We start by introducing some notation. Let $y_1, y_2, \ldots, y_n$ be multivariate observations from samples $i = 1, 2, \ldots, n$, which can be features from genomics, neuroimaging, or any type of multivariate data. Let $D = (d_{ij})_{n \times n}$ denote the sample dissimilarity matrix computed on these observations $y_i$, where

$d_{ij} = d(y_i, y_j)$ and $d$ is a chosen dissimilarity function. We define the doubly-centered dissimilarity matrix $G = (I - \mathbf{1}\mathbf{1}^T)A(I - \mathbf{1}\mathbf{1}^T)$ where $A = \left(-\frac{1}{2}d_{ij}^2\right)_{n \times n}$.

## 2.1 Adjusted principal coordinates analysis

We first review a traditional dimension reduction method called principal coordinates analysis (PCoA), which finds coordinates in Euclidean space that optimally preserve dissimiliarites from the original space. The classical solution finds these coordinates via eigendecomposition of $G$ [15]. Decomposing $G = U\Lambda U^T$, these coordinates are given by $Z = U\Lambda^{1/2}$. Under Euclidean dissimiliarities, the principal coordinates $Z$ preserve the exact distances from the original space. If the original dissimilarities are non-Euclidean, $Z$ may contain imaginary coordinates. Adding a constant to every pairwise dissimilarity can produce coordinates in Euclidean space [16].

In adjusted principal coordinates analysis (aPCoA), a linear model is used to remove the effect of nuisance covariates from the principal coordinates [5]. Let $X$ be an $n \times p$ design matrix of nuisance covariates with corresponding projection matrix $H = X(X^TX)^{-1}X^T$. aPCoA first assumes a linear model with respect to the nuisance covariates in the space of principal coordinates. Then, aPCoA regresses out these nuisance covariates yielding covariate-adjusted coordinates $E = (I - H)Z$. Their corresponding dissimilarity matrix is referred to as the covariate-adjusted dissimilarity matrix $\Delta = EE^T = (I - H)G(I - H)$. aPCoA has been demonstrated to yield data visualizations while accounting for confounding by examining the first two covariate-adjusted coordinates [5].

Under certain conditions, aPCoA assumes the same model as multivariate distance matrix regression (MDMR; [17–19]). When the original dissimilarity metric is Euclidean, MDMR is equivalent to testing a linear association between principal coordinates and the covariates using a psuedo-F statistic [17]. For other dissimilarity metrics, MDMR tests using the covariate-adjusted dissimilarity matrix from aPCoA as the denominator of the psuedo-F statistic. This equivalence suggests that aPCoA targets nuisance associations detectable via MDMR, which are known to include complex nonlinear associations in the original space [17–19].

## 2.2 Partial embeddings

We develop the partial embedding (PARE) framework for deconfounded dimension reduction by removing the effect of confounders from pairwise dissimilarities. We use the covariate-adjusted dissimilarity matrix $\Delta = (\delta_{ij})_{n \times n} = (I - H)G(I - H)$ as an input into any distance-based dimension reduction method, which include $t$-SNE [8] and UMAP [9] among others. This framework extends aPCoA by enabling removal of nuisance effects from a broader class of dimension reduction methods. For example, UMAP defines affinities based on dissimilarity metrics $d$ as $v_{ij} = v_{j|i} + v_{i|j} - v_{j|i}v_{i|j}$ where

$$v_{j|i} = \exp[(-d(y_i, y_j) - \rho_i)/\tau_i]$$

and $\rho_i$ are the dissimilarity to the nearest neighbor of $y_i$ and $\tau_i$ are normalizing factors computed based on dissimilarities among a chosen number of nearest neighbors. A PARE for UMAP can be formulated via the adjusted affinities

$$v_{j|i}^{\text{PARE}} = \exp[(-\delta_{ij} - \rho_i)/\tau_i].$$

For Euclidean distances, dimension reduction methods can instead take the principal coordinates as inputs. As examples, we highlight how $t$-SNE and UMAP can be equivalently formulated in terms of principal coordinates. $t$-SNE measures similarity in the original space as

affinities $p_{ij} = \frac{p_{j|i}+p_{i|j}}{2n}$ under a Gaussian kernel where

$$p_{j|i} = \frac{\exp(-\| y_i - y_j \|^2/2\sigma_i^2)}{\sum_{k\neq i}\exp(-\| y_i - y_k \|^2/2\sigma_i^2)},$$

$\|\cdot\|$ is the Euclidean norm, and $\sigma_i$ are chosen to yield a specified perplexity value for each observation. UMAP using Euclidean distances defines similarities using a locally adaptive exponential kernel as $v_{ij} = v_{j|i} + v_{i|j} - v_{j|i}v_{i|j}$ where

$$v_{j|i} = \exp[(- \| y_i - y_j \| -\rho_i)/\tau_i].$$

Let $z_i$ denote the principal coordinate vector for observation $i$. For Euclidean distances in the original space, the principal coordinates have identical pairwise distances such that $\|z_i - z_j\| = \|y_i - y_j\|$. Then the $t$-SNE and UMAP affinities can be written in terms of principal coordinates as

$$p_{j|i} = \frac{\exp(-\| z_i - z_j \|^2/2\sigma_i^2)}{\sum_{k\neq i}\exp(-\| z_i - z_k \|^2/2\sigma_i^2)},$$

$$v_{j|i} = \exp[(- \| z_i - z_j \| -\rho_i)/\tau_i].$$

We develop PAREs for any dimension reduction method based on Euclidean distances by instead taking adjusted principal coordinates $e_i = (I - H)z_i$ as input data. These adjusted coordinates preserve dissimilarities while removing unwanted effects due to the nuisance covariates $X$. We outline the steps in constructing PAREs using Euclidean distances below:

1. Obtain principal coordinates $Z$ from the original data from the Euclidean distance matrix $D$ as described in subsection 2.1.

2. Using a linear model, residualize $Z$ with respect to nuisance covariates $X$ to obtain adjusted coordinates $E = (I - H)Z$, where $H = X(X^TX)^{-1}X^T$.

3. Input the adjusted coordinates $E$ to any dimension reduction method based on Euclidean distances.

Obtaining these adjusted coordinates only requires eigendecomposition of the original dissimilarity matrix followed by residualization using a linear model. Both steps are implemented via multiple packages in R, Python, MATLAB, and other programming languages.

For our investigation, we apply our PARE framework to $t$-SNE and UMAP using Euclidean distances to develop p-$t$-SNE and p-UMAP. We use R (version 4.1.1) implementations for $t$-SNE and UMAP in the packages Rtsne (version 0.15) and umap (version 0.2.7.0). Throughout our applications, we choose the perplexity as 10 for $t$-SNE and the number of nearest neighbors as 15 for UMAP.

## 3 Materials

### 3.1 Ethics statement

Written consent was obtained from all subjects involved in this study. Genomic data from [20] (8569 cells, inDrop protocol), [21] (1050 cells, SMARTer), [22] (2122 cells, CEL-Seq2), and [23] (2133 cells, SMART-Seq2) all were approved by their corresponding IRBs. For the NAIMS traveling subject data, we received written informed consent from all participants, which was approved by the University of Pennsylvania's institutional review board (IRB).

ADNI obtained all written IRB approvals and met all ethical standards in the collection of data. The following are the ethics committees and IRB boards that provided approval. The Ethics committees/institutional review boards that approved the ADNI study are: Albany Medical Center Committee on Research Involving Human Subjects Institutional Review Board, Boston University Medical Campus and Boston Medical Center Institutional Review Board, Butler Hospital Institutional Review Board, Cleveland Clinic Institutional Review Board, Columbia University Medical Center Institutional Review Board, Duke University Health System Institutional Review Board, Emory Institutional Review Board, Georgetown University Institutional Review Board, Health Sciences Institutional Review Board, Houston Methodist Institutional Review Board, Howard University Office of Regulatory Research Compliance, Icahn School of Medicine at Mount Sinai Program for the Protection of Human Subjects, Indiana University Institutional Review Board, Institutional Review Board of Baylor College of Medicine, Jewish General Hospital Research Ethics Board, Johns Hopkins Medicine Institutional Review Board, Lifespan—Rhode Island Hospital Institutional Review Board, Mayo Clinic Institutional Review Board, Mount Sinai Medical Center Institutional Review Board, Nathan Kline Institute for Psychiatric Research & Rockland Psychiatric Center Institutional Review Board, New York University Langone Medical Center School of Medicine Institutional Review Board, Northwestern University Institutional Review Board, Oregon Health and Science University Institutional Review Board, Partners Human Research Committee Research Ethics, Board Sunnybrook Health Sciences Centre, Roper St. Francis Healthcare Institutional Review Board, Rush University Medical Center Institutional Review Board, St. Joseph's Phoenix Institutional Review Board, Stanford Institutional Review Board, The Ohio State University Institutional Review Board, University Hospitals Cleveland Medical Center Institutional Review Board, University of Alabama Office of the IRB, University of British Columbia Research Ethics Board, University of California Davis Institutional Review Board Administration, University of California Los Angeles Office of the Human Research Protection Program, University of California San Diego Human Research Protections Program, University of California San Francisco Human Research Protection Program, University of Iowa Institutional Review Board, University of Kansas Medical Center Human Subjects Committee, University of Kentucky Medical Institutional Review Board, University of Michigan Medical School Institutional Review Board, University of Pennsylvania Institutional Review Board, University of Pittsburgh Institutional Review Board, University of Rochester Research Subjects Review Board, University of South Florida Institutional Review Board, University of Southern, California Institutional Review Board, UT Southwestern Institution Review Board, VA Long Beach Healthcare System Institutional Review Board, Vanderbilt University Medical Center Institutional Review Board, Wake Forest School of Medicine Institutional Review Board, Washington University School of Medicine Institutional Review Board, Western Institutional Review Board, Western University Health Sciences Research Ethics Board, and Yale University Institutional Review Board.

## 3.2 Human pancreatic cell scRNA-seq data

We apply PAREs to human pancreatic cell scRNA-seq data to remove batch and donor effects from data collected across four separate studies with varying number of cells and RNA-seq protocol. We include RNA-seq data from [20] (8569 cells, inDrop protocol), [21] (1050 cells, SMARTer), [22] (2122 cells, CEL-Seq2), and [23] (2133 cells, SMART-Seq2). We treat each study as a separate batch and treat each donor as distinct across studies. We follow a preprocessing pipeline proposed in [24]. First, we use Scran (release 3.15) in R to perform log-normalization and selection of highly variable genes (HVGs) using the counts data from each

study. Genes that are not present in all four studies were removed from further evaluation. We then perform normalization by computing size factors across pools of cells, then obtaining factors for each cell via a deconvolution approach [24]. Within each study, locally weighted scatterplot smoothing (LOESS) is applied to model the mean-variance relationship among genes. We then use a weighted arithmetic mean of mean and variance statistics across studies to select 2,000 HVGs. We remove cells labelled as "unclear", "none", "unclassified" or "co-expression". After preprocessing, our human pancreatic cell dataset is comprised of 13,369 cells with four batches, 26 donors, and 13 cell types.

We apply p-$t$-SNE and p-UMAP to remove batch effect or donor effects in the embeddings. We compare p-$t$-SNE to other methods that simultaneously remove confounding effects and obtain lower-dimensional representations including projected $t$-SNE for batch correction (BC-$t$-SNE; [6]) and Adjustment for Confounding factors using Principal Coordinate Analysis (AC-PCoA; [25]), proposed in their original paper. For BC-$t$-SNE, we set the perplexity at 10 to be similar to our p-$t$-SNE approach. For AC-PCoA, we compare AC-PCoA followed by $t$-SNE to p-$t$-SNE with respect to study across varying numbers of coordinates for both methods. Due to computational complexity of AC-PCoA, we perform these comparisons in a dataset consisting of human pancreatic cell scRNA-seq data from three donors: donor ACCG268 from [21] (136 cells), donor AZ from [23] (63 cells), and donor D28 from [22] (181 cells).

In this setting, our method is also comparable to two-stage approaches where batch correction is performed prior to dimension reduction. These approaches include Combatting Batch Effects (ComBat; [26]), fastMNN [27], and Harmony [28] which are all widely employed and have been compared recently [29–31]. For our analyses, we compare p-$t$-SNE and p-UMAP to performing ComBat, fastMNN, or Harmony followed by $t$-SNE or UMAP. ComBat is run using the R implementation in the ComBatHarmonization package (github.com/Jfortin1/ComBatHarmonization). fastMNN is run with default settings using the batchelor package version 1.19.1 in R. Harmony is performed with default settings on the top 20 principal components using the harmony package version 1.2.0.

We compare our methods visually and numerically using the local inverse Simpson's index (LISI; [28]) and the average sillouette width (ASW; [32]). Recent work has demonstrated that these metrics can effectively evaluate performance of modern batch correction methods [29, 30]. For measuring integration of cells across batches, we compute LISI for batch (bLISI), which captures the effective number of batches in a local neighborhood around each cell. We also examine LISI computed for cell type (cLISI), which captures the number of neighboring cell types and decreases as the separation between cell types increases. We compute bLISI and cLISI across a range of perplexity values, which capture different neighborhood sizes. Additionally, we compute the ASW for batch (bASW), which is a distance-based statistic that compares the average distance of one cell to cells of its own batch versus other batches. Lower bASW indicates better batch correction performance. For cell type, we similarly compute ASW for cell type (cASW) where higher cASW indicates better separation between cell types.

### 3.3 ADNI cortical thickness dataset

We apply PAREs to brain cortical thickness data from the Alzheimer's Disease Neuroimaging Initiative (ADNI) to distinguish technical and biological variability. The data for this study consist of baseline scans which are processed using the ANTs longitudinal single-subject template pipeline [33] with code available on GitHub (github.com/ntustison/CrossLong). The ADNI study obtained written informed consent from all participants. Further details for this preprocessing pipeline can be found in [34].

The full sample consists of 505 subjects, 213 of whom are imaged on scanners manufactured by Siemens, 70 by Philips, and 222 by GE. The sample has a mean age of 75.3 (SD 6.70) and is comprised of 278 (55%) males, 115 (22.8%) Alzheimer's disease (AD) patients, 239 (47.3%) late mild cognitive impairment (LMCI), and 151 (29.9%) cognitively normal (CN) individuals. We apply p-$t$-SNE and p-UMAP to separate effects of diagnosis and scanner. Quantitatively, we evaluate our methods using the ASW and LISI for manufacturer and diagnosis. Higher manufacturer LISI (mLISI) and lower manufacturer ASW (mASW) indicate better batch adjustment performance. Lower diagnosis LISI (dLISI) and higher diagnosis ASW (dASW) indicate better separation between diagnosis groups.

### 3.4 NAIMS traveling subjects study

To examine if PAREs can identify technical variability not visible in $t$-SNE and UMAP embeddings, we apply our PAREs to a study of patients with multiple sclerosis (MS) with multiple scan-rescan images across four different sites in the North American Imaging in Multiple Sclerosis (NAIMS) Cooperative. These sites include the University of Pennsylvania (Penn), the Brigham and Women's Hospital (BWH), the National Institutes of Health (NIH), and the Johns Hopkins University (Hopkins). Nine of the eleven participants are scanned at all four study centers. The mean age of our 11 participants (4 male, 7 female) at time of enrollment was 38 (range 29–47).

A standardized high-resoution 3-tesla (3T) MRI brain scan protocol developed by the NAIMS Cooperative was performed at each site [35]. Images were acquired on Siemens Skyra (BWH, NIH), Siemens Prisma (Penn), and Philips Achieva (Hopkins) scanners. Each participant had two scans acquired on the same day at each visit to the study center.

Prior to automated segmentation, images undergo bias correction via nonuniform intensity normalization (N4ITK; [36]) and FLAIR images are rigidly aligned to the corresponding T1-weighted image within a given scan session. Brain extraction is performed using Multi-Atlas Skull Stripping (MASS; [37]) and intensity normalization is performed using White-Stripe [38]. White matter and gray matter volumes are estimated using Joint Label Fusion (JLF; [39]), a segmentation method that leverages information from several atlases via weighted voting. These JLF volumes are used as inputs into $t$-SNE, UMAP, and our PARE methods. We use PAREs to identify scanner effects independently of within-subject similarities. We additionally evaluate our methods using the ASW and LISI for site and subject. Lower site LISI (bLISI) and higher site ASW (bASW) indicate greater separation between sites. Lower subject LISI (sLISI) and higher subject ASW (sASW) indicate greater separation between subjects.

## 4 Results

### 4.1 Case study 1: Human pancreatic cells

We first apply PAREs to analyze scRNA-seq data from human pancreatic cells across four published studies, treated as separate batches [20–23]. We observe clear batch effects in the original $t$-SNE and UMAP visualizations along with a lack of integration among several cell types (Fig 1). Applying p-$t$-SNE and p-UMAP to remove batch effects considerably reduces separation by batch both visually and numerically (Fig 2), as measured by increases in bLISI and decreases in bASW for p-UMAP (UMAP median bASW = 0.22 for UMAP, p-UMAP w.r.t. study median bASW = =0.06). Remaining batch differences can partially be explained by donor effects, since PAREs with respect to donor show greater visual integration across batches and higher bLISI. PAREs also achieve greater distinction of cell types as measured by decreases in cell type LISI (cLISI, Fig 2). Comparing p-$t$-SNE to the existing projected $t$-SNE

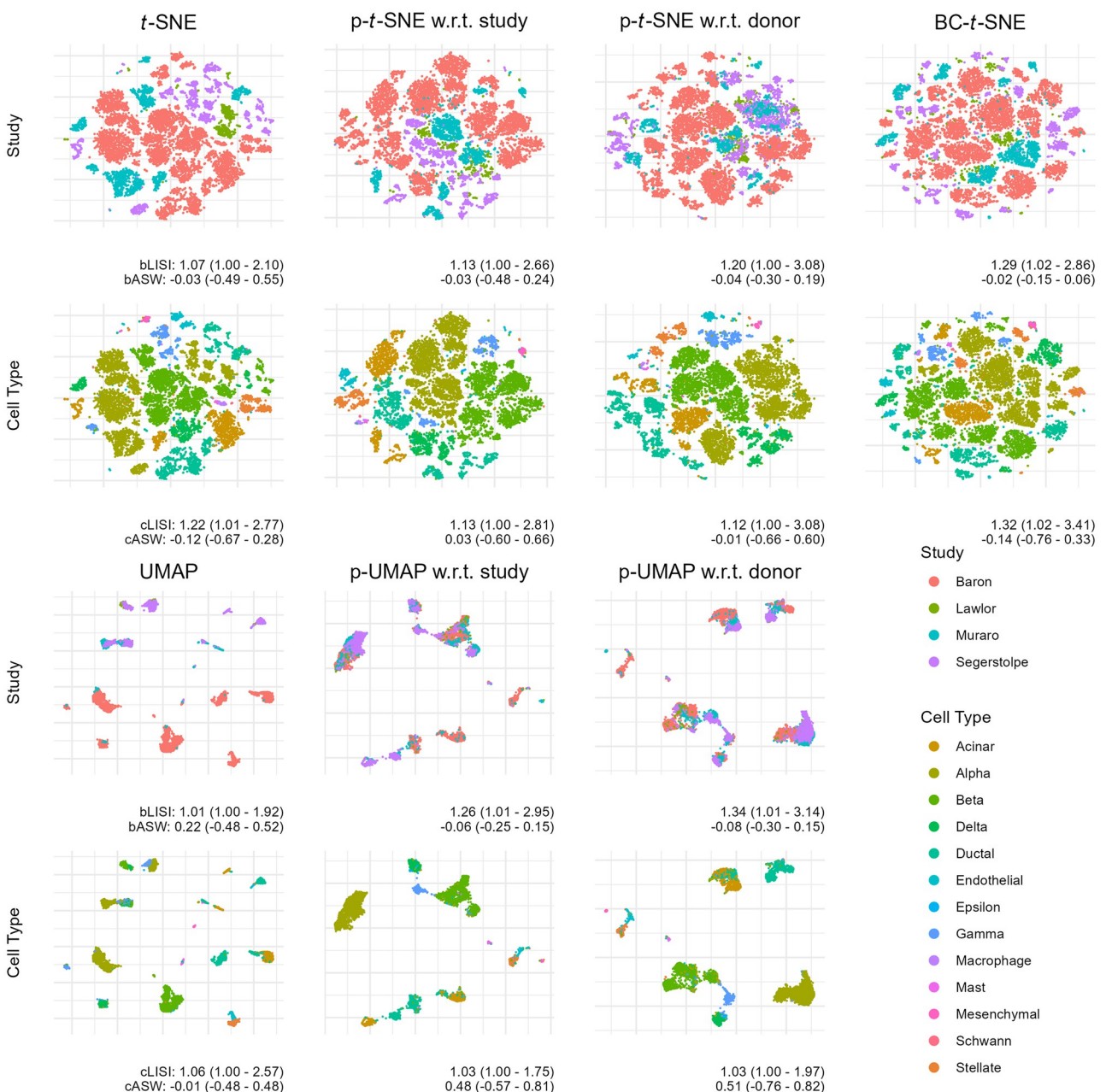

**Fig 1. Partial embeddings remove batch and donor effects in single-cell RNA-sequencing measurements aggregated from four studies.**
Embeddings and partial embeddings are compared in data from 13,369 human pancreatic cells. The original counts data is log-normalized and reduced to 2,000 highly variable genes. Local inverse Simpson's index and average silhouette width are computed for each cell for batch (bLISI, bASW) and cell type (cLISI, cASW) with the median, 2.5% quantile, and 97.5% quantile shown. Higher bLISI and lower bASW indicate greater integration across batches. Lower cLISI and higher cASW indicate greater separation between cell types. Partial t-SNE (p-t-SNE) and partial UMAP (p-UMAP) adjust for either batch or donor effects. We compare our new methodology to the existing projected t-SNE for batch correction (BC-t-SNE). All t-SNE embeddings have a perplexity of 10 and UMAP embeddings use 15 nearest neighbors.

for batch correction (BC-t-SNE; [6]), we find that BC-t-SNE achieves greater batch integration but obscures important biological patterns that separate cell types (median cLISI increases from 1.22 in t-SNE to 1.32 in BC-t-SNE). Compared to batch correction using ComBat, fastMNN, or Harmony followed by dimension reduction, p-t-SNE and p-UMAP w.r.t. study

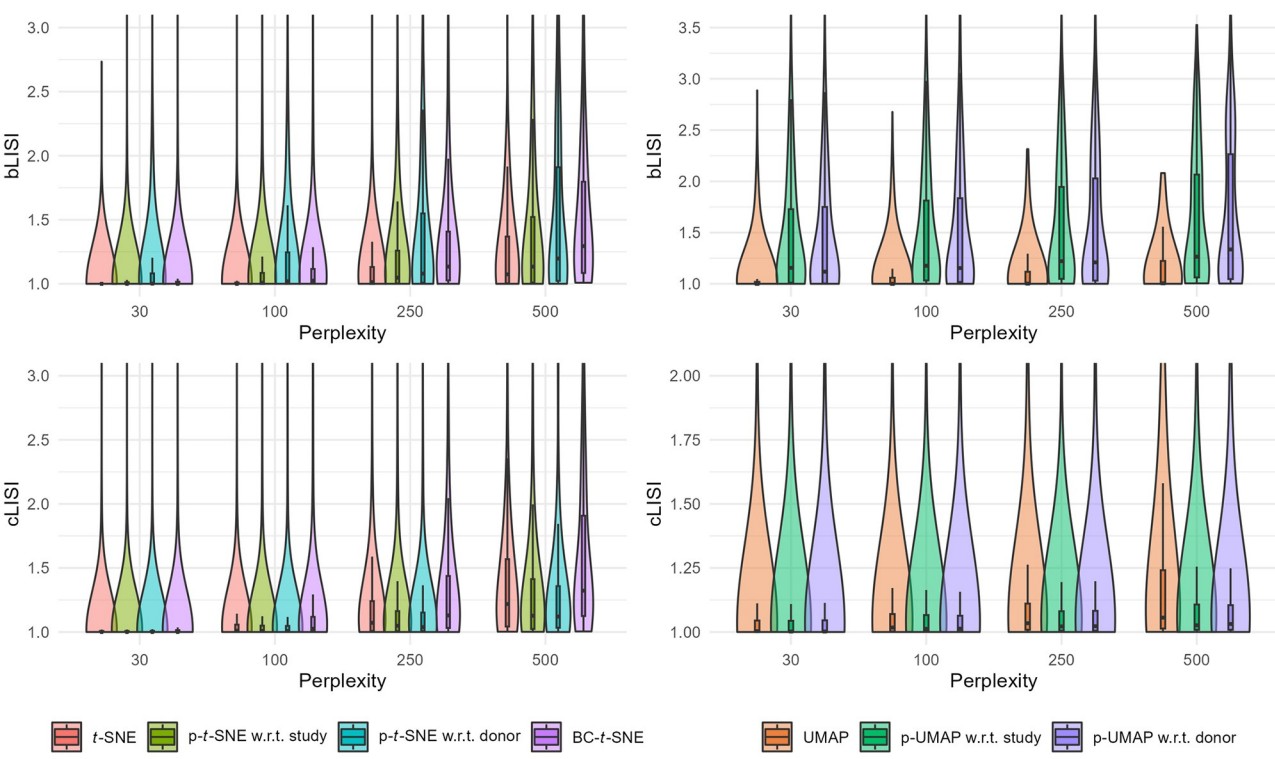

**Fig 2. Quantitative comparisons via local inverse Simpson's index show that partial embeddings outperform across multiple perplexity values.**
Local inverse Simpson's index for batch (bLISI) and cell type (cLISI) are computed on log-normalized single-cell RNA-sequencing data from 13,369 human pancreatic cells across four studies. The original embeddings and partial embeddings are compared across perplexity values, which capture different neighborhood sizes around each cell. Higher bLISI corresponds to better batch adjustment performance and lower cLISI indicates greater separation between cell types. We compare our new methodology to the existing projected *t*-SNE for batch correction (BC-*t*-SNE).

demonstrate comparable performance but fastMNN+*t*-SNE and fastMNN+UMAP outperform other methods (**Figs A and B in** S1 Text). In a subset of three donors, we additionally compare p-*t*-SNE to AC-PCoA+*t*-SNE, showing comparable performance across varying numbers of coordinates (**Fig C in** S1 Text). We finally show in Fig 3 that effective results can be achieved by computing a subset of principal coordinates, which is less computationally intensive. In scRNA-seq data, we demonstrate that PAREs can effectively isolate biological variability from unwanted sampling effects in scRNA-seq data.

## 4.2 Case study 2: Brain cortical thickness

We next apply PAREs to brain cortical structure measurements to separate biological effects from scanner-related artifacts in the Alzheimer's Disease Neuroimaging Initiative (ADNI) study. In the ADNI study, researchers previously identified diagnosis-related atrophy in cortical structure and notable batch effects due to differences in scanner properties across study sites [34, 40]. In Fig 4A, we observe that the original embeddings display both of these effects. However, the confounding scanner effects result in the overlap among images acquired on a Siemens scanner and those from patients with an Alzheimer's disease (AD) diagnosis. To specifically investigate differences between people with and without AD, we demonstrate that PAREs adjusted for scanner manufacturer maintain diagnosis effects while obscuring scanner influence (Fig 4A). PAREs are also used to examine scanner effects without the influence of diagnosis effects, highlighting known differences among scanners in the ADNI study.

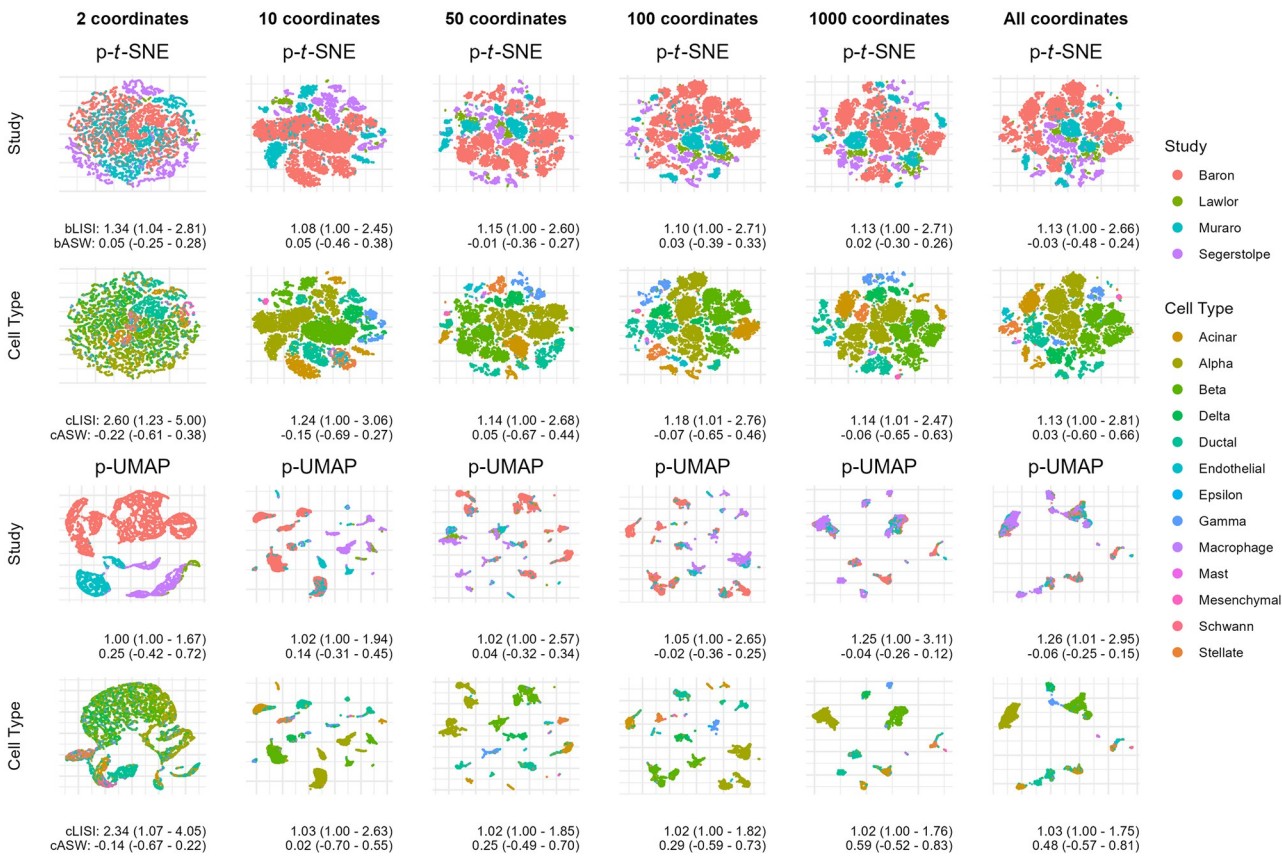

**Fig 3. Partial embeddings estimated using fewer principal coordinates.** Partial embeddings of single-cell RNA-sequencing are estimated using varying numbers of principal coordinates. Each dimension reduction method takes a subset of adjusted or unadjusted principal coordinates. The dimension of this subset is varied across figure columns. Partial *t*-SNE (p-*t*-SNE) and partial UMAP (p-UMAP) adjust for batches.

## 4.3 Case study 3: Traveling subject brain volumetric data

Finally, as a proof of concept, we use PAREs to identify scanner effects in brain white matter and gray matter volumes collected as part of a multi-site traveling subjects study of multiple sclerosis (MS). The study involves eleven MS patients with multiple scans across four major imaging centers. We include Siemens images with distortion correction, which was designed to reduce differences with the Philips scanner at Johns Hopkins University (Hopkins). Original *t*-SNE and UMAP embeddings clearly separate white matter and gray matter volumetric measurements by subject regardless of site (Fig 4B). However, these original embeddings do not capture other types of variability, including potential site effects. We apply p-*t*-SNE and p-UMAP to remove subject effects and discover deviation of images acquired on the Philips scanner at Hopkins from those acquired on Siemens scanners at other sites (Fig 4B). Here, we show that PAREs can identify technical variability in neuroimaging measures that could not be detected in the original embeddings.

## 5 Discussion and conclusion

We propose the PARE framework, which extends any distance-based dimension reduction method to adjust for confounders. Our analyses demonstrate that PAREs can be used to target specific patterns in high-dimensional data by removal of confounders. We demonstrate that

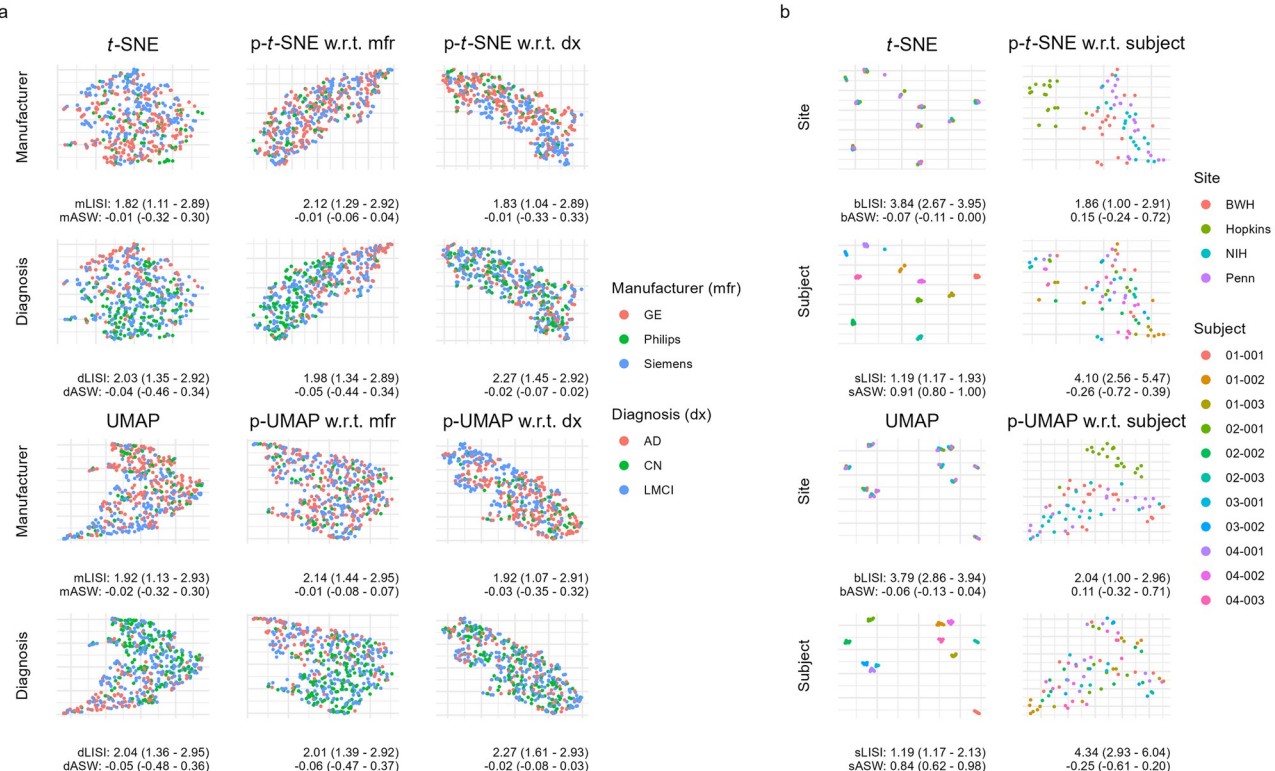

**Fig 4. Partial embeddings separate biological and technical variability in brain cortical thickness measurements (a) and regional volumes (b) across two multi-site neuroimaging studies.** Embeddings and partial embeddings are compared visually and numerically using the local inverse Simpson's index (LISI) and average silhouette width (ASW). **(a)** visualizes cortical thickness data from the Alzheimer's Disease Neuroimaging Initiative, from which we include 505 participants. These participants are diagnosed as cognitively normal (CN), having late mild cognitive impairment (LMCI), or having Alzheimer's disease (AD). Participants are acquired across scanners with three distinct manufacturers. Higher manufacturer LISI (mLISI) and lower manufacturer ASW (mASW) indicate better batch adjustment performance. Lower diagnosis LISI (dLISI) and higher diagnosis ASW (dASW) indicate better separation between diagnosis groups. **(b)** shows results from a traveling subjects study of eleven multiple sclerosis (MS) patients with multiple images across four study sites. The Hopkins site uses a Philips scanner while the three other sites use Siemens scanners. Lower site LISI (bLISI) and higher site ASW (bASW) indicate greater separation between sites. Lower subject LISI (sLISI) and higher subject ASW (sASW) indicate greater separation between subjects.

our proposed PAREs are able to remove batch effects in scRNA-seq exploration, emphasize diagnosis-related changes in brain cortical structure, and identify scanner effects in brain volumetric measurements. For dimension reduction based on Euclidean distances, our PARE framework relies solely on PCoA and linear regression, which are both widely available and computationally efficient. While we only investigate PAREs built on *t*-SNE and UMAP, this framework can be readily applied to a broad class of dimension reduction methods based on distances from the original space.

We note several limitations of the current manuscript. First of all, PARE is limited to exploratory data visualization and cannot address confounding effects in further analyses. We do not recommend PAREs to be used in place of the original data in downstream analyses or claim that PAREs preserve all data properties, since distance-based embeddings in general have been demonstrated to distort properties of the original data [41]. For removal of confounding from downstream analyses, the use of an adjusted dissimilarity matrix has been previously explored and demonstrated to reduce confounding in nonparametric analyses while preserving data properties [25]. As an alternative for batch correction, harmonization methods have been extensively used to reduce batch-related confounding in visualization and

downstream analyses [29–31]. Compared to harmonization followed by dimension reduction, we find that PAREs can underperform fastMNN followed by $t$-SNE or UMAP. However, PAREs have several use cases beyond batch adjustment since PAREs can handle continuous confounding variables and multiple confounders. Throughout our case studies, we perform computational evaluations using the neighborhood-based LISI and distance-based ASW statistics, which are two separate approaches for evaluating separation between categories. In our scRNA-seq analyses, the LISI and ASW generally provide congruent results. However, we note that there are other statistics which will be considered in future studies, including several metrics that have been compared to the LISI and ASW across a larger number of scRNA-seq datasets [29, 30].

The PARE framework opens several new directions for methodological development. Future investigations can examine how the PARE framework performs for extensions of methods not considered in this article. For Euclidean distances, PAREs are constructed from the residuals of a linear model, but other models including linear mixed models and general additive models could also be considered for longitudinal and non-linear effects. Extensions of the PARE framework can also readily incorporate multiple complex data types by integrating at the level of principal coordinates, suggested in a recently proposed multimodal regression model [42]. Furthermore, PAREs can be extended to examine data types independently of one another by projection of dissimilarity matrices [43]. In summary, the PARE framework is able to remove nuisance effects in any distance-based dimension reduction tool. Our framework enables discovery of notable patterns in complex high-dimensional data and introduces a foundation for future methodological research.

## Supporting information

**S1 Text. The Supplementary Materials contain supplementary figures associated with the manuscript.**
(PDF)

## Author Contributions

**Conceptualization:** Andrew A. Chen.

**Data curation:** Kelly Clark, Blake E. Dewey, Anna DuVal, Nicole Pellegrini, Govind Nair, Youmna Jalkh, Samar Khalil, Jon Zurawski, Peter A. Calabresi, Daniel S. Reich, Rohit Bakshi, Russell T. Shinohara.

**Formal analysis:** Andrew A. Chen.

**Funding acquisition:** Haochang Shou, Russell T. Shinohara.

**Methodology:** Andrew A. Chen.

**Software:** Andrew A. Chen.

**Supervision:** Haochang Shou, Russell T. Shinohara.

**Writing – original draft:** Andrew A. Chen.

**Writing – review & editing:** Kelly Clark, Blake E. Dewey, Anna DuVal, Nicole Pellegrini, Govind Nair, Youmna Jalkh, Samar Khalil, Jon Zurawski, Peter A. Calabresi, Daniel S. Reich, Rohit Bakshi, Haochang Shou, Russell T. Shinohara.

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
