## [Decision Letter · Decision Letter 0]

20 Dec 2023

Dear Assistant Professor Chen,

Thank you very much for submitting your manuscript "PARE: A framework for removal of confounding effects from any distance-based dimension reduction method" for consideration at PLOS Computational Biology.

As with all papers reviewed by the journal, your manuscript was reviewed by members of the editorial board and by several independent reviewers. In light of the reviews (below this email), we would like to invite the resubmission of a significantly-revised version that takes into account the reviewers' comments. Please pay attention especially to the comments by the second reviewer, and ensure that the code can be viewed by the reviewer.

We cannot make any decision about publication until we have seen the revised manuscript and your response to the reviewers' comments. Your revised manuscript is also likely to be sent to reviewers for further evaluation.

Sincerely,

Matthias Helge Hennig, Ph.D.

Academic Editor

PLOS Computational Biology

Marieke van Vugt

Section Editor

PLOS Computational Biology

Reviewer's Responses to Questions

**Comments to the Authors:**

Reviewer #1: This paper proposes an approach for the removal of confounding sources of variation prior to visualization using dimension reduction methods including t-SNE and UMAP. The method is clearly described and illustrated on three real-world data sets.

Comments:

* The approach for removing the confounding variation assumes a linear relation between the confounders and a Euclidean representation of the data. There is a newer method AC-PCoA (Wang et al. 2022) that allows for non-linear effects of confounders. Is it possible to utilize this framework to construct the covariate-adjusted dissimilarities?

* An alternative approach to removing confounding effects in visualization would be to apply a recommended batch correction method for the given data type (e.g., Harmony or Seurat for single cell data), and then apply t-SNE or UMAP to the resulting data. How does this approach compare to the proposed method?

* The presentation of some of the figures could be improved. In particular, the violin plots in Figure 2 have the same heights across all methods, making it hard to identify potentially interesting differences in performance across methods.

* Some other figures rely on the reader to carefully squint at many colored dots (e.g., Figure 4). Could a quantitative summary be used here as well to support the qualitative conclusions?

Minor comments/typos:

* p. 2 : “These methods has…” -> “These methods have…”

* p. 4: “Define the …” -> “We define the …”

Reviewer #2: The review is uploaded as an attachment. I would like to emphasise that I can only recommend this work for publication, if both major concerns outlined in the review are addressed and resolved in a satisfactory manner.

**Have the authors made all data and (if applicable) computational code underlying the findings in their manuscript fully available?**

Reviewer #1: Yes

Reviewer #2: **No: **I cannot seem to be able to find the code for PARE online.

PLOS authors have the option to publish the peer review history of their article (what does this mean?). If published, this will include your full peer review and any attached files.

Reviewer #1: No

Reviewer #2: No
---

## [Decision Letter · Decision Letter 1]

10 Jun 2024

Dear Assistant Professor Chen,

We are pleased to inform you that your manuscript 'PARE: A framework for removal of confounding effects from any distance-based dimension reduction method' has been provisionally accepted for publication in PLOS Computational Biology.

Best regards,

Matthias Helge Hennig, Ph.D.

Academic Editor

PLOS Computational Biology

Marieke van Vugt

Section Editor

PLOS Computational Biology

Reviewer's Responses to Questions

**Comments to the Authors:**

Reviewer #1: I am satisfied that my previous review comments have been addressed.

Reviewer #2: The authors have addressed my comments and have made the corresponding adjustments to the manuscript.

**Have the authors made all data and (if applicable) computational code underlying the findings in their manuscript fully available?**

Reviewer #1: Yes

Reviewer #2: Yes

PLOS authors have the option to publish the peer review history of their article (what does this mean?). If published, this will include your full peer review and any attached files.

Reviewer #1: No

Reviewer #2: No

---

## [Editor Report · Acceptance letter]

1 Jul 2024

PCOMPBIOL-D-23-01698R1 

PARE: A framework for removal of confounding effects from any distance-based dimension reduction method

Dear Dr Chen,

I am pleased to inform you that your manuscript has been formally accepted for publication in PLOS Computational Biology. Your manuscript is now with our production department and you will be notified of the publication date in due course.

With kind regards,

Olena Szabo
